# HSF1-Activated Non-Coding Stress Response: Satellite lncRNAs and Beyond, an Emerging Story with a Complex Scenario

**DOI:** 10.3390/genes13040597

**Published:** 2022-03-27

**Authors:** Claire Vourc’h, Solenne Dufour, Kalina Timcheva, Daphné Seigneurin-Berny, André Verdel

**Affiliations:** 1Université de Grenoble Alpes (UGA), 38700 La Tronche, France; 2Institute for Advanced Biosciences (IAB), Centre de Recherche UGA/Inserm U 1209/CNRS UMR 5309, Site Santé-Allée des Alpes, 38700 La Tronche, France; solenne.guerin@univ-grenoble-alpes.fr (S.D.); kalina.timcheva@kuleuven.be (K.T.); daphne.berny@univ-grenoble-alpes.fr (D.S.-B.)

**Keywords:** HSF1, lncRNA, *SATIII*, *eRNA*, *SINE*, *TERRA*, *NEAT1*

## Abstract

In eukaryotes, the heat shock response is orchestrated by a transcription factor named Heat Shock Factor 1 (HSF1). HSF1 is mostly characterized for its role in activating the expression of a repertoire of protein-coding genes, including the heat shock protein (HSP) genes. Remarkably, a growing set of reports indicate that, upon heat shock, HSF1 also targets various non-coding regions of the genome. Focusing primarily on mammals, this review aims at reporting the identity of the non-coding genomic sites directly bound by HSF1, and at describing the molecular function of the long non-coding RNAs (lncRNAs) produced in response to HSF1 binding. The described non-coding genomic targets of HSF1 are pericentric Satellite DNA repeats, (sub)telomeric DNA repeats, Short Interspersed Nuclear Element (SINE) repeats, transcriptionally active enhancers and the NEAT1 gene. This diverse set of non-coding genomic sites, which already appears to be an integral part of the cellular response to stress, may only represent the first of many. Thus, the study of the evolutionary conserved heat stress response has the potential to emerge as a powerful cellular context to study lncRNAs, produced from repeated or unique DNA regions, with a regulatory function that is often well-documented but a mode of action that remains largely unknown.

## 1. Introduction

The cellular response mediated by Heat Shock transcription Factor 1 (HSF1) is involved in a large number of biological processes and is widespread within the eukaryotic kingdom. Initially described in the context of the cellular response to thermal stress, the involvement of HSF1 has been rapidly extended to a large number of other environmental and cellular stresses and physio-pathological contexts [1,2,3,4]. Genes encoding Heat Shock Proteins (HSPs) are major effectors of the stress response and of HSF1 [5]. HSPs maintain protein homeostasis in a disturbed physiological context through their action on protein folding [6]. In addition, non-HSP protein-coding genes targeted by HSF1 have been identified with roles in the regulation of apoptosis, cellular defense against Reactive Oxygen Species (ROS), and cell membrane and chromatin organization [7,8,9].

Today, the emerging view is that, in addition to the well-known response of inducing the expression of specific protein-coding genes, HSF1 also mediates a non-coding response to stress (non-coding referring to the fact that the transcripts generated here do not code for proteins). This non-coding response may be involved in the rapid and coordinated reshaping of protein-coding gene expression through a potentially large range of mechanisms [10]. Indeed, besides the HSP and other proteins, diverse long non-coding RNAs (lncRNAs) have been identified, mostly over the last two decades, as direct genomic targets of HSF1. These transcripts include Satellite III transcripts (*SATIII*) emanating from pericentric heterochromatin (Figure 1A) [11,12] as potential effectors of the heat shock response. The biological relevance of this HSF1-mediated non-coding response, and the mechanisms involving the regulatory lncRNAs produced, are only starting to be understood.

This review focuses primarily on the known mammalian lncRNA effectors of the HSF1 response, with the aim of providing an updated view of their role and mode of action in the HSF1-dependent stress response. The first part reports and briefly describes the lncRNAs whose production is directly activated by HSF1 (Figure 1A,B). These lncRNAs are the *SATIII* and *TERRA* transcripts expressed from DNA repeats in pericentric [11,12], and (sub)telomeric heterochromatin [13,14], respectively, the Short Interspersed Nuclear Elements (*SINEs*) transcripts that are expressed from repetitive sequences scattered throughout the genome [15,16], enhancer RNAs (*eRNAs*) [8,9] and the two lncRNA isoforms, *NEAT1.1* and *NEAT1.2*, expressed from the single-copy *NEAT1* gene [17]. The second part of the review presents the emerging mechanisms by which these lncRNAs may contribute to HSF1-dependent stress-response.

## 2. The Genomic Noncoding Sequences Transcribed under the Direct Control of HSF1

Several lines of evidence indicate that HSF1 directly controls expression of a set of genes not coding for proteins. These assumptions rely on computational genomic analyses allowing an identification of putative HSF1-binding sites known as Heat Shock Elements (HSE), Chromatin Immuno-Precipitation (ChIP) to evaluate the binding of HSF1 to their promoter and, finally, comparative analysis of their levels of expression in cells expressing HSF1 or not. We will provide here a short description of these lncRNAs and the findings indicating that the genes encoding these lncRNAs are directly upregulated by HSF1 (Table 1).

### 2.1. Satellite lncRNAs

Satellite DNA sequences, present at pericentric regions, are enriched in tandem repeats organized in a “head-to-tail” orientation.

In humans, the size and sequence of pericentric regions varies between chromosomes. The majority of them contain, as a major component, a single family of simple repeated satellite (SAT) sequences formed by the classical *satellites I, II and III*. Of these, *SATIII* are characterized by the high frequency of the *sat3* motif [19,20,21] (also see Table 2, associated references [20,21,22,23,24,25,26,27,28,29]).

More recent sequencing analysis of human satellite regions led to a more refined identification of chromosome-specific satellite 2 and 3 motifs with the identification of 10 subfamilies, each displaying specific chromosome-specific distributions [27]. In mouse, pericentric regions of ~6 Mb are made of a repetition of 234 bp of AT-rich sequences, known as the major satellite motif [23]. Pericentric regions are enriched in methylated DNA, [30] and other repressive marks, such as H3K9me2/3 and Heterochromatin Protein 1 (HP1) [19,31,32]. In addition, repression at pericentric regions also involves a BRCA1-mediated H2A monoubiquitinylation [33,34]. The very recent complete sequencing of human pericentric regions reveals that, as in mouse, these sequences cover several megabases, much longer in size than initially estimated [35]. For example, the largest region containing *SATIII* repeats, found within the chromosome 9 pericentric heterochromatin, is around 27 megabases long [35].

#### HSF1 in the Control of Human *SATIII* Repeats’ Expression

Human *SATIII* repeats have been found to be up-regulated by HSF1 [11,12]. In heat-shocked cells, a pericentric region of chromosome 9 (locus 9q12) [11,12,36] and, more recently, of chromosome Y (locus Yq12) [37] were both characterized as primary targets of HSF1. However, additional transcription sites form on pericentric regions of other chromosomes in cells overexpressing HSF1 and in cells under more severe stress, suggesting that low-affinity HSF1-binding sites also exist on other pericentric regions [38]. The total number of HSF1-binding sites at these loci is unknown. However, an analysis of genomic sequences from the 9q12 locus revealed the presence of putative Heat Shock Elements (HSEs) over stretches of *SATIII* repetitive motifs (our unpublished observation). *SATIII* are transcribed by RNAPII into a pool of *SATIII* lncRNAs ranging in size from 2 Kb to 5 Kb and beyond [11,12]. *SATIII* DNA repeats are predominantly transcribed in one orientation (enriched in the pentanucleotide repeat motif GGAAU). Interestingly, a slight increase in the level of *SATIII* lncRNAs in an antisense ((AUUCC)n-rich) orientation was also observed in stressed cells [11,12,39], but there is no evidence to date for a direct role of HSF1 in up-regulating antisense *SATIII* lncRNAs.

In addition, different cellular contexts involving pericentric satellite lncRNAs’ accumulation have also been described, but for which a direct role of HSF1 has not been demonstrated so far. For instance, in mouse, an accumulation of major satellite RNA from pericentric heterochromatin has been reported during late G1/S and mitotic stages of the cell cycle [40] and during early development [41]. Transcripts produced from pericentric regions are also present in both mouse [42] and human [43,44] testis. Finally, an accumulation of satellite transcripts, including *SATIII*, was also observed in a variety of human tumor tissues [44,45,46].

### 2.2. Telomeric Repeat Containing RNA (TERRA) lncRNAs

Telomeric heterochromatin consists of tandem repeats of the TTAGGG hexanucleotide motif [47,48]. These repeats form the “telomeric tract” that spans several kilobases at the ends of chromosomes. In stressed cells, telomeric and subtelomeric regions are transcribed by RNAPII into heterogeneous lncRNAs called TElomeric Repeat-containing RNA (*TERRA*), which range in size from 100 bases to >100 Kb [49,50]. *TERRA* production is initiated in specific regions of the chromosome located approximately 1 Kb upstream of the telomeric tract. *TERRA*s therefore contain chromosome-specific sequences in addition to a common telomeric tract [50]. *TERRA*s have been found to be expressed from several human chromosome ends [51,52,53,54] and *TERRA* are associated with telomeric DNA repeats [55,56].

#### HSF1 in the Control of Human *TERRA*s’ Expression

The first observation that heat shock increases transcription at telomeres was reported in Chironomus thummi [57] then confirmed in mouse [56] and human [14]. The direct implication of HSF1 in the up-regulation of *TERRA* was then reported in human [13]. In human cells, HSF1 binding and HSF1-dependent transcriptional activation of telomeric repeats was observed specifically at telomeres containing HSEs, which are only present in 40% of all sub-telomeres [13].

### 2.3. Short Interspersed Nuclear Element (SINE)-Containing lncRNAs

*SINE*s are present as thousands of copies interspersed in the genome of mouse and higher primates [58,59]. In mouse, they include *B1*- and *B2*-like elements. *B1*-like elements are ancestrally derived from the *7SL* gene, which is a component of the signal-recognition particle (SRP) ribonucleoprotein complex with a role in protein synthesis [15]. *B2*-like elements are derived from tRNA precursors and are present in mouse and rabbit. In human, B1-like elements are known as *Alu* elements [60]. *Alu* elements are approximately 280 bases in length, and are formed by two tandem repeats separated by a central A-tract [15]. Sequences encoding *Alu* elements have an internal RNA polymerase III (RNAPIII) promoter that has the potential to initiate transcription of the *Alu* sequence. Since *Alu* sequences lack a termination site for transcription, they are transcribed into RNAs of heterogenous length depending on the location of nearby transcription terminator [61]. Under normal growth conditions, the internal RNAPIII promoter is necessary but not sufficient for transcription, and *Alu* sequences are not transcribed, unless they are fortuitously positioned near appropriate strong enhancers [62]. Furthermore, the repressive histone H3 lysine 9 trimethylation (H3K9me3) mark seems to silence the majority of *SINE* sequences, such that only a very low level of *SINE* RNAs is produced from the internal RNAPIII promoter, at dispersed loci in mouse and human [63]. However, because of their ubiquitous interspersion within RNAPII transcribed units, *SINE*s are also abundantly expressed as short-lived RNA as part of intronic sequences of nuclear transcripts [60,64].

#### HSF1 in the Control of SINEs’ Expression

An increase in *Alu* elements and *Alu* retrotransposition events has been described in response to genotoxic stresses [65]. Importantly, in humans, the targeting of *Alu* sequences by HSF1, inducing their expression upon heat stress, was demonstrated by means of primer extension upon exposure to mild heat shock [16]. Likewise, HSEs were identified at conserved sites in different Alu subfamilies within *Alu* sequences [66].

### 2.4. Enhancer RNAs (eRNAs)

Enhancer RNAs represent an emerging class of ncRNAs produced at distal Transcription Regulatory Elements (dTREs), which are defined as genomic regions, localized outside of gene promoters, that increase transcription of neighboring genes [67]. dTREs are transcribed bi-directionally by RNAPII within domains preferentially enriched in H3K4me1 and H3K27ac [68,69]. Based on their size (in the 50 bp-2 Kb sized-range), *eRNA*s belong mostly to the category of lncRNAs.

#### HSF1 in the Control of *eRNA*s’ Expression

Interestingly, two recent papers report HSF1 binding to enhancers that produce *eRNA*s in both mouse and human cells [8,9]. In a model of human myeloid/erythroid leukemia K562 cells submitted or not to heat shock, precision run-on and sequencing (PRO-seq) experiments identified thousands of dTREs that include enhancer regions, in both unstressed and stressed cells. In this study, more than 5000 dTREs were found to be significantly up-regulated upon heat-shock [8]. Noticeably, a repertoire of around 400 dTREs observed in heat-shocked cells and encompassing HSF1-binding sites showed a higher density of RNAPII and a higher level of histone acetylation [8], suggesting that they are direct targets of HSF1.

### 2.5. Nuclear-Enriched Abundant Transcript 1 (NEAT1)

The human NEAT1/Men ε/β gene codes for two overlapping 3.7 Kb and 22.3 Kb lncRNA isoforms called *NEAT1-1* and *NEAT1-2*, respectively, which are generated from the same promoter [70]. In mammalian tissues and cells, the NEAT1-2 isoform localizes in specific nuclear structures called paraspeckles [71,72,73]. Paraspeckles are spherical membrane-less ribonucleoprotein structures formed by the association of *NEAT1* lncRNAs and various proteins. *NEAT1-2* RNA is essential for the formation and maintenance of paraspeckles [74,75]. *NEAT1-1* lncRNA is not essential for paraspeckles formation but enhances paraspeckles formation when *NEAT1-2* is present [74]. Unlike *NEAT1-2*, *NEAT1-1* can be found outside of paraspeckles in tissue culture cells, further suggesting that it may have different biological functions than *NEAT1-2* [76]. The *NEAT1* gene sequence is not well conserved between human and mouse but in both organisms, a common strong propensity for long-range intramolecular base-pairing was recently found to contribute to paraspeckle scaffolding [77]. An average of 5–20 paraspeckles (0.5–1 µm in diameter) are present in unstressed human cells with the exception of human embryonic stem cells (hESC), the only mammalian cell type reported to be devoid of paraspeckles [78].

#### HSF1 in the Control of Human *NEAT1* Expression

Recently, a direct HSF1-dependent up-regulation of *NEAT1-1* and *NEAT1-2* lncRNAs was described in human breast cancer MCF7 cells submitted to a mild 30-min heat shock at 43 °C followed by a recovery period of 2 h at 37 °C [17]. Interestingly, the Heat Shock Element (HSE)-binding sites found at the *NEAT1* promoter were also present within *NEAT1* promoters in other eukaryotes [17].

## 3. Molecular Functions of HSF1-Driven Production of lncRNAs in Response to Heat Stress

In this second part of the review, we present how this non-coding fraction of the genome directly targeted by HSF1 for transcription activation may act in the heat shock response. What are the common features, if any, of the lncRNAs identified so far? With the exception of *NEAT1* lncRNAs, they are transcribed from a large number of genomic repeats, either distributed along all chromosomes (*Alu* RNAs, *eRNA*s), or at more specific and relatively large genomic loci (*TERRA*, *SATIII* lncRNAs). In addition, two of them (*TERRA*, *SATIII* lncRNAs) consist of a high density of tandem repeats units of a short nucleic acid motif. At least two of them (*SATIII* lncRNAs, *NEAT1*) are essential scaffolding elements for the formation of sub-nuclear and membrane-less compartments [79,80] (see Table 1).

### 3.1. LncRNA-Dependent Nuclear Relocation of Factors Involved in the Control of Transcription

In human stressed cells, *SATIII* lncRNAs remain at their site of transcription, even several hours after mild stress exposure (30 min exposure at 43 °C). The nuclear structures that form at sites of *SATIII* transcription are referred to as nuclear Stress Bodies (or nSBs) [81]. This name embraces the notion that these structures are enriched in various transcription and co-transcription factors (including HSF1, HATs and BET proteins) and RNA-binding factors [79,82,83,84].

The accumulation of transcriptional and co-transcriptional factors at nSBs may contribute to a global deacetylation of the two core histone proteins H3 and H4 [85,86] detected by western blot [85] or immunofluorescence performed on stressed cells [11,12]. Indeed, massive recruitment of HSF1 and its associated Histone Acetyl Transferases (HATs), especially at the 27-megabases-long 9q12 pericentric locus that is highly enriched in *SATIII* DNA repeats, could represent an efficient way of promoting a genome-wide reduction in acetylation and, thus, controlling genome-wide expression. Several HATs (Gcn5, CBP/p300 and Tip60) [11,87], and chromatin readers, such as Bromodomain extra-terminal domain (BET)-containing proteins [84,87], are recruited to nSBs in an HSF1-dependent manner [11,87] (Figure 2A). Furthermore, stress-induced global deacetylation of core histones throughout the whole genome is temporally coupled with increased acetylation of core histones at the *SATIII*-enriched 9q12 locus [7,87,88]. This deacetylation process is rapid and impacts lysine residues present in the *N*-terminal or histone fold regions. Histone acetylation loosens histone–DNA interactions by neutralizing the positive charges of lysine and also provides docking sites for histone-binding proteins that play an important role in the recruitment or activity of essential transcription factors [89]. While a knock-down of *SATIII* lncRNAs partially releases the heat-induced repression of a series of genes, overexpression of ectopic *SATIII* lncRNAs triggers gene repression in both unstressed and stressed cells and results in a significant decrease in the survival of stressed cells [88].

### 3.2. LncRNA-Dependent Nuclear Relocation of Factors Involved in the Control of Splicing

Similarly, the specific impact of heat stress on the alteration of the intranuclear distribution of components of the splicing machinery has been known for a long time and illustrated in a number of selected target RNAs and organisms [90,91,92,93,94], with heat stress found to block constitutive pre-mRNA splicing but also to affect the regulation of alternative splicing [82]. The stress-induced dephosphorylation of SRSF10 (SRP38) plays an important role in the heat-induced inhibition of splicing [95]. Dephosphorylated serine- and arginine-rich protein SRSF10 (SRP38) binds to U1 snRNP-associated protein U1 70K, and prevents the interaction of U1 70K with other SR proteins [96].

In addition, other serine-rich (*SR*) *proteins*, including (SF2/ASF-SRSF1), Srp30c (SRSF9), 9G8 (SRSF7) and heterogeneous nuclear ribonucleoprotein (hnRNPs-hnRNPA1)-interacting proteins (HAP/SAF-B and hnRNPM), with essential roles in constitutive and alternative pre-mRNA splicing, are massively recruited to HSF1-dependent nSBs in human stressed cells. The recruitment of these alternative splicing factors to nSBs is selective and proteins with constitutive roles in splicing are not recruited [97]. Different mechanisms may be involved in the recruitment of these factors at nSBs. For example, while the interaction of SAFB, SRSF1, SRSF7, SRSF9 and HNRNPM with *SATIII* lncRNAs is potentially direct [98], involving, in the case of SRSF1, a defined RNA-binding domain [12,99], other proteins such as HAP are recruited through protein/protein interactions [100]. Furthermore, the deposition of N^6^-methyladenosine (m^6^A) RNA modification on *SATIII* lncRNAs may also recruit proteins such as YTHDC1 [101] (and unpublished data from our team). These different modes of recruitment may explain the differences in the kinetics of recruitment of these proteins to nSBs, relative to the timing of accumulation of *SATIII* lncRNAs [79,81].

It is possible that recruitment of these splicing proteins is intended to regulate the splicing of *SATIII* lncRNAs [102]. However, no direct evidence exists that *SATIII* transcripts are subjected to a complete splicing reaction [99]. Instead, an alternative view could be that, through a selective recruitment of splicing factors, *SATIII* lncRNAs may impact the balance between SR and hnRNP proteins, and thereby regulate the alternative splicing of many protein-coding transcripts containing introns [79,82]. In favor of this possibility, a more complete picture of the impact of heat shock on splicing, based on transcriptome-wide approaches, has emerged recently with studies performed on mouse [103] and human cells [84,98,101]. In both models, a widespread inhibition of alternative splicing has been observed in heat-shocked cells, mostly characterized by a high percentage of intron-retention [84,98,103]. Interestingly, intron retention is even greater in human heat-shocked cells’ knock-down for *SATIII* expression [84]. This suggests that the accumulation of *SATIII* lncRNAs protects pre-mRNAs from splicing alterations [84].

In addition, during recovery from stress, CDC-like kinase 1 (CLK1) is recruited to nSBs. *SATIII* lncRNAs, appear to act as platforms that can concentrate SRSFs in a small fraction of the nucleus for their efficient and timely phosphorylation by CLK1, in a manner that promotes a rapid adaptation of gene expression, through the control of intron excision, following thermal stress exposure [98].

As with *SATIII*, *NEAT1* lncRNAs also seem to play a role in RNA processing. Indeed, *NEAT1* lncRNAs have been found to anchor the DROSHA-DGCR8 microprocessor complex to paraspeckles via a pri-miR-612 (and possibly additional) stem-loop structure(s) present within the *NEAT1* sequence [104]. Other proteins, including NONO (P54NRB) and PSF (or-SFPQ), both involved in pri-miRNA processing and identified as key component of paraspeckles, may assist in the transport and processing of other pri-miRNAs to the paraspeckles [80,104]. Up-regulation of miRNAs resulting from HSF1-induced accumulation of *NEAT1* could also be involved in the control of gene expression in stressed cells (64). Indeed, a reduction in almost all expressed miRNAs was observed in cells lacking *NEAT1* [104] and could be related to the up-regulation of HSP genes, reported in stressed cells knocked down for *NEAT1* [17] (Figure 2A). The exact implication of *NEAT1* up-regulation in terms of miRNA-mediated changes in gene expression in heat-shocked cells, however, remains to be determined.

What might be the advantage of regulating these RNA-based processes through the sequestration of key players? Transient sequestration of RNA processing factors and RNA regulators could be less metabolically demanding than the degradation of these factors followed by their synthesis and could allow rapid adaptation to stressful environments. Such regulation might actually be conserved through evolution. In Drosophila, the heat shock RNA omega (*hsrω*) gene, encoding *hsrω* lncRNAs, is upregulated in heat-shocked cells [105,106,107]. Although transcribed from a single-copy gene, they also accumulate within nuclear structures called omega speckles [108]. Similarly to nSBs, omega speckles act as dynamic storage sites for various hnRNPs and other RNA processing and regulatory proteins [108], raising the attractive hypothesis that *SATIII* and *hsrω* lncRNAs may represent functional analogs in the regulation of RNA processing and function upon stress [109,110].

### 3.3. Interaction of lncRNAs with the Transcriptional Machinery

The cellular response to stress is characterized by major and rapid changes in gene transcription, as demonstrated in mouse [9] and human [8], with Precision nuclear Run-On sequencing (PRO-seq). This technique allows a fine profiling of the dynamic regulation of the transcriptionally engaged RNA polymerase transcription across the genome [111]. Likewise, in human K562 myeloid erythroid cells, exposure to moderate thermal shock induces a down-regulation of a large fraction of protein-coding gene expression (estimated at 36% of the total number of genes analyzed) and, conversely, an up-regulation of a much smaller fraction of protein-coding genes (4.6%) [8].

HSF1-dependent transcriptional activation is primarily modulated at the level of promoter-proximal pause and release, with HSF1 increasing RNAPII release from promoter-proximal pause [8,9] and dissociation of the 330-nucleotide-long *7SK* RNA from positive Transcription Elongation Factor b (pTEFb) [112]. Different types of mechanisms involving a direct impact of lncRNAs on the transcriptional machinery have been described in stressed cells. LncRNAs have been shown to play a role in the 3D-structuration of functional promoter regions in the control of transcription initiation and elongation.

The production of Enhancer RNA, at HSF1-binding sites located in the enhancer regions of stress-induced genes, is thought to bring together transcription factors present in the distal promoter region’s elements and in the core promoter [113] or to maintain transcription-binding sites in an open chromatin configuration [114,115] (Figure 2B).

Additionally, *SINE* RNAs have been found to impact RNAPII-dependent transcription at different levels. *SINE* RNAs have been shown to affect RNAPII-recruitment at the preinitiation complex. Observations in human and mouse heat-shocked cells revealed that SINE transcripts (produced by RNAPIII and not RNAPII) establish a direct interaction with RNAPII [116,117]. More specifically, *B2* RNA (mouse) and *Alu* RNA (human) disrupt contacts between RNAPII and promoter DNA [118,119] (Figure 2C, left). The interaction between SINEs RNA and RNAPII prevents the formation of preinitiation complexes [118]. Furthermore, still in stressed cells, *B2* RNA can also control transcriptional elongation levels by another mechanism. Indeed, B2 RNA-binding sites have been identified at many genic and intergenic targets, including at heat-responsive genes in mouse [120], just proximal to the sites of RNAPII pausing [9]. EZH2 is a member of the polycomb complex that is involved in gene repression and which catalyzes the addition of methyl groups to H3K27. Upon heat shock, EZH2 possesses an additional RNA processing function and is able to trigger rapid cleavage of *B2* RNA, allowing RNAPII elongation (Figure 2C, right) [120]. Since a binding of *B2* RNA to chromatin is not detected in the gene body of *HSP* genes, up-regulated genes controlled by *B2* RNA and up-regulated genes controlled by HSF1 likely represent different subsets of genes [120].

### 3.4. LncRNA-Dependent Recruitment of Repressive Complexes

Non-coding transcripts produced from pericentric heterochromatin are known to be essential actors in the maintenance of heterochromatin in mouse [121,122]. Major Satellite lncRNAs (which are thought to be functionally equivalent to human *SATIII* lncRNAs) are essential in setting up pericentric heterochromatin and chromocenter formation in early embryonic development and play an important role in the deposition of the repressive H3K9me3 histone mark to pericentric heterochromatin regions at that stage [41,123]. In this context, although not demonstrated yet, the hypothesis that *SATIII* lncRNAs may participate in the reformation of pericentric heterochromatin in the recovery period from stress is, therefore, attractive at a stage when repressive marks need to be restored.

In the fission yeast *Schizosaccharomyces pombe*, a model of choice to study heterochromatin, RNA of pericentric origin is required to initiate and maintain heterochromatin. In this case, pericentric lncRNAs form double-stranded RNAs that are diced into small interfering RNAs (siRNAs, which are around 20 bases long) and also serve as a platform to recruit a repressive complex. So far, we have no evidence that RNAi mechanisms involving *SATIII* RNA also occur in early development or in the recovery period of stress. In addition, the precise biological context in which the RNAi pathway participates or not in the maintenance of heterochromatin at centromeric regions is still unclear in higher eukaryotes. In unstressed mouse ES cells, small centromeric and pericentric double-stranded RNA (dsRNA) molecules, with predominant signals at 150 nt and 25–30 nt, are only detected in DICER-expressing cells [124]. Still in mouse ES cells, an accumulation of transcripts of pericentric origin is observed in DICER−/− cells [124,125], thus suggesting a DICER-dependent silencing mechanism for centromeric heterochromatin [125]. In human cells, the accumulation of human centromeric and pericentric specific transcripts, ranging in size from 20–30 nt up to several Kb, was observed in a chicken–human hybrid cell line model [126], while siRNA-targeting DICER A and B isoforms were not found to lead to an accumulation of *SATIII* RNA in human HeLa cells [44]. In HeLa cells, an accumulation of *SATIII* RNAs molecules ranging in size from 25 to 75 nt (detected as a ladder with a regular 5 nt increment) was also described in the recovery period from stress [127]. While in human cells, no evidence exists that DICER is involved in the accumulation of *SATIII* RNA in stressed cells, a rapid accumulation of small 21–30 nt siRNAs originating from major DNA components of centromeric and pericentric regions (TCAST) was reported in the beetle *Tribolium castaneum* submitted to a transient heat shock stress [128]. This accumulation correlates with the enrichment of H3K9me2 and H3K9me3 at pericentric heterochromatin following stress, suggesting a possible implication of TCAST 21–30 nt siRNAs in the reformation of silent chromatin marks [128]. Based on these data, a clarification regarding the possible implication of siRNAs in heterochromatin reformation in mammals remains necessary. Differences in the mechanisms of heterochromatin formation exist among eukaryotes and what is true in flies may not be necessarily transposable to mammals.

Another non-mutually exclusive possibility is that heterochromatin reformation after stress may involve the use of lncRNAs without the involvement of a process such as RNAi. These lncRNAs would then solely act as platforms to recruit proteins necessary for the formation of heterochromatin, similar to what has been described for the *Xist* (X inactivating specific transcripts) lncRNAs that coat the inactive X chromosome within the Barr body in female mammals [129]. In the same line, TERRAs could be exerting the same recruiting platform function [51,52,130]. Chromatin at telomeres is enriched with the repressive histone marks of H3K9me3, and TERRAs have been detected in an enzymatic complex catalyzing H3K27 trimethylation at the telomeres. H3K27me3 is required in turn for the establishment of H3K9me3, H4K20me3 and HP1 binding at the telomeres (Figure 2D) [131,132,133].

The discovery of the accumulation of SINE transcripts in heat-shocked cells also raised the possibility that these transcripts may regulate gene expression in *cis* [66]. Indeed, *Alu* elements containing HSE-binding sites are frequently positioned downstream of the transcription start site of genes, suggesting that HSF1 may contribute to the down-regulation of sense transcripts in heat-shocked cells by promoting the production of antisense lncRNAs. In support of this hypothesis, a knockdown of antisense transcripts produced at several of these loci was found to increase the expression of the corresponding sense RNA [66] (Figure 2E).

### 3.5. LncRNA-Dependent Control of Genome Integrity

A source of HSF1-driven genomic instability could result from the stress-induced alteration of centromeric regions (including centromeric regions and pericentric heterochromatin regions) caused by the transcription of *SATIII* DNA repeats. Evidence exists that lncRNAs of centromeric origin are important players in kinetochore complex stability, kinetochore assembly and spindle assembly and that these RNAs represent an integral part of centromere identity [134,135,136,137,138,139,140,141]. However, a tight control of the lncRNA expression of centromeric origin is required for normal functional centromeres in unstressed and stressed cells. Indeed, both an overexpression [142,143] or a silencing [138] of centromeric transcripts leads to chromosome segregation defects. In Drosophila, centromeric *SATIII* lncRNAs that span several megabases on the X chromosome are also required for correct localization of the centromere-defining proteins CENP-A and CENP-C, as well as for outer kinetochore proteins [144], and a knock down of Drosophila *SATIII* transcripts causes mitotic defects [144].

The need for a better characterization of the impact of stress-induced *SATIII* lncRNAs accumulation on the structure and function of centromeres in human cells appears to be an important issue. We have no evidence so far that, as observed in Drosophila, human *SATIII* lncRNAs play a role in the stability of kinetochore. However, a recent publication reports that a transient prometaphase arrest occurs in stressed HeLa cells, the duration of which correlates with the level of *SATIII* lncRNAs accumulation [127]. Noticeably, HeLa cells that go through mitosis before the level of *SATIII* lncRNAs has returned to normal show chromosome-9-specific segregation defects [127]. In this context, stress-induced *SATIII* lncRNAs accumulation appears to be a double-edged sword, as *SATIII* lncRNAs’ accumulation may be part of a checkpoint, which is eventually bypassed in cancer cells, and thus provide a source of chromosomal instability. The impact of heat stress on chromosome instability may also result from an alteration of DNA replication. Indeed, several publications point to the impact of *SATIII* RNAs on DNA replication. For example, loss of BRCA1 in breast tumor correlates with an upregulation of *SATIII* RNA resulting in chromosome instability [34]. Overexpression of *SATIII* RNAs in breast cancer deficient for BRCA1 was found to destabilize replication forks through the interaction of *SATIII* RNA with the members of the complex required for DNA replication composed of BRCA1-interacting protein [145]. The titration of members of the BRCA1 complex away from stalled replication forks results in the formation of RNA–DNA hybrids and in the accumulation of γH2AX (a marker for DNA damage) at the replication forks [145].

Finally, *SATIII* lncRNAs appear as interesting therapeutic markers in cancer therapy. Together with heat-shocked cells, cancer cells overexpressing *SATIII* RNA as a consequence of DNA hypomethylation at pericentric heterochromatic regions are resistant to etoposide, a topoisomerase IIa (TOP2a) inhibitor. The interaction of TOP2a with *SATIII* RNA at nSBs protects TOP2A against interaction with etoposide, preventing increased DNA damage in etoposide-treated cells [46].

The possibility that the HSF1-induced accumulation of lncRNAs may play a role in genome protection is perhaps best illustrated in the case of TERRA. In unstressed cells, *TERRA* transcripts are thought to play a role in telomere stability, telomere replication and the telomeric DNA damage response [51]. In many eukaryotes, telomeres are protected from double-stranded breaks and end-fusions by a protein complex, the shelterin complex [146]. The interaction of *TERRA* with the shelterin components TRF1 and TRF2 (Telomere Repeat binding Factors 1 and 2) contributes to telomere length homeostasis and protection [146], and may also anchor *TERRA* to chromosome ends, sustaining the enzymatic activities of TERRA-binding factors at telomeres [147]. Yet, many questions remain regarding the source of *TERRA* production, its distribution and its role [133,148].

Interestingly, an HSF1-dependent up-regulation of TERRA has been observed at specific telomeres in stressed cells [13,14,56]. Heat shock induces the partial loss of TRF2 at the telomeres [13], which is also dependent on HSF1. TRF2 is a member of the Shelterin complex, which specifically prevents telomeres’ recognition as double-stranded DNA breaks [51,52], by repressing the activation of the ATM (Ataxia Telangiectasia Mutated) kinase signaling pathway [54]. Moreover, TRF2 protects telomeres against end-fusions elicited by the non-homologous end-joining pathway [54] and assists replication fork progression through pericentric regions, by preventing the accumulation of G-quadruplexes-like structures [149]. While a loss of HSF1 does not prevent TRF2 dissociation, a higher number of telomere dysfunction-induced foci (TIF) was detected in HSF1 knock-down stressed cells [13]. TIF correspond to regions enriched in the phosphorylated protein, γH2AX, mostly known as a biomarker of double DNA-strand breaks but also possibly enriched at stress-induced arrested replication forks [150,151]. The up-regulation of *TERRA* correlates with the recruitment of LSD1, a lysine-specific demethylase, at telomeres [54]. LSD1 directly interacts with *TERRA* and Mre11, a H3K4 methyltransferase subunit of the MRE11/RAD50/NBS1 (MRN) repair complex that may promote telomere fusions [54]. Whatever is the signal that triggers γH2AX enrichment at telomeres, this enrichment reveals a critical role for HSF1 in telomere protection in stressed cells, which is possibly related with its role on TERRA upregulation [13]. Finally, a protective role for TERRA may also rely on its ability to recruit, as mentioned previously [129,130,131], repressive chromatin complexes to telomeres, and thereby promote the formation of more compact chromatin with greater resistance to stress-induced damage (Figure 2D).

Other stress-induced lncRNAs may also have a direct impact on genome stability. For example, *Alu* sequences have been found to undergo retrotransposition upon stress [65]. Unlike Long Interspersed Nuclear Elements (*LINE*), another class of interspersed repetitive sequences, also present in thousands of copies in AT-rich and gene-poor genomic regions, *SINE*s are not capable of retrotransposition by themselves. However, with the help of *LINE* sequences, which provide them with the necessary elements for transposition, transposition of *Alu* elements has been reported. Retrotransposition of *Alu* elements was observed in human cells submitted to etoposide [65] and one cannot exclude that this type of event could occur in the same way in heat-shocked cells. Heat shock was also found to promote transposition events in other organisms including plants [152] and insects [153]. In *Arabidopsis thaliana*, the heat-shock-induced transposition of ONSEN, a *Ty1/copia-like* retrotransposon [154], and of its homolog that is also present in *Vigna angularis* [152] has been observed, representing a mechanism that may help the organisms to adapt to new environmental conditions [155,156]. At least in *Arabidopsis thaliana*, this transposition is mediated by the Heat Shock transcription Factor A2 (HSFA2), a functional equivalent of HSF1 [154]. However, this stress-induced transposition has only been reported in plants deficient in RNA-directed DNA methylation (RdDM) (a mechanism involving double-stranded 21–24 bases long small interfering RNAs (siRNAs) that guide methylation to silence retro-transposons) [154]. Although the retro-transposition of pericentromeric human *sat2* motifs has been found to expand locally and genome-wide in primary human colon cancer samples [157], there is no evidence that heat shock is able to trigger a retrotransposition of *SINE* and of *SATIII* sequences in human cells.

## 4. Conclusions

An emerging view of the HSF1-mediated non-coding response to stress is that it plays a global role in controlling gene expression by regulating transcriptional and post-transcriptional processes. The HSF1-mediated non-coding response should also be considered for its impact on genome stability. The non-coding fractions of eukaryotic genomes have long been neglected in genomic and gene expression studies. While only a small number of lncRNAs have been considered so far as direct effectors of HSF1 in stressed cells, it already appears that these RNAs are important actors of the heat shock response. The functional analysis of some of these RNAs is still hampered by the difficulty to efficiently knock-down nuclear RNAs and by the fact that they may be transcribed from multiple genomic loci, e.g., in genomic regions that are highly enriched in Satellite DNA repeats. The development of new tools such as CRISPR-Cas9 to knock out genomic sequences [158] and CRISPR-Cas13 to specifically target and knock-down nuclear and chromatin-associated transcripts represent promising new approaches to better study regulatory lncRNAs, especially in response to stress [159]. An important aspect will be to better define the different physiological and pathophysiological contexts of their accumulation. In this perspective, the study of Satellite lncRNAs clearly illustrates the difficulties and black holes. Different classes of Satellite transcripts from pericentric origin accumulate in both cancer tissues and heat-stressed cells. However, we do not know to what extent these Satellite sequences are structurally and functionally equivalent or not, in both cellular contexts. More generally, since these sequences are not conserved through evolution, a comparative analysis of their roles in different eukaryotes is often difficult and the molecular and functional characterization of their mode of action is an exciting challenge. Other aspects including the possibility that the accumulation of these lncRNAs may be linked to the regulation of the epi-transcriptome [101,160], a new dimension in the control of gene expression as well as translation, must also be considered. Finally, growing attention is also being paid to the contribution of lncRNAs in nucleating, maintaining and regulating the formation of membrane-less nuclear compartments through liquid–liquid phase separations (LLPS) [161,162,163]. The solidification of HSF1 foci within a sustained stressing environment was found to have a negative impact on *HSP* gene expression [164]. Trapping of HSF1 within nSBs was proposed to mark the cells with excessive proteotoxic damage and turn a reversible response into a signal to apoptosis with irreversible outcome [164]. Other types of HSF1-dependent stress-induced non-coding transcription, such as *NEAT1* transcription [165], may serve as effective entities to orchestrate and reshape gene expression. In addition, at least one microRNA (miRNA) has already been identified as a direct target of HSF1 in a human cervical cancer cell line [166]. Knowing that miRNA can regulate several mRNA targets, an extended identification of the microRNA genes targeted by HSF1 and the impact of these miRNAs in the HSF1-dependent regulation of gene expression certainly represents another important perspective [167,168].

The extraordinary diversity of the new regulators, issued from the emerging population of ncRNAs accumulating in response to a stress and the HSF1-dependent activation of their transcription, such as *SATIII* lncRNAs, brings a new dimension to the orchestration of the cellular response to stress by HSF1. Finally, HSF1 overexpression and activation was also observed in cancer cells while the knock-out of HSF1 was found to protect mice from tumorigenesis induced by mutations of the RAS oncogene or by a hot spot mutation in the tumor suppressor p53 [169]. It is, therefore, tempting to consider that HSF1-dependent production of lncRNAs may also produce effectors that play key roles in the protumoral effect of the HSF1 signaling pathway.

## Figures and Tables

**Figure 1 genes-13-00597-f001:**
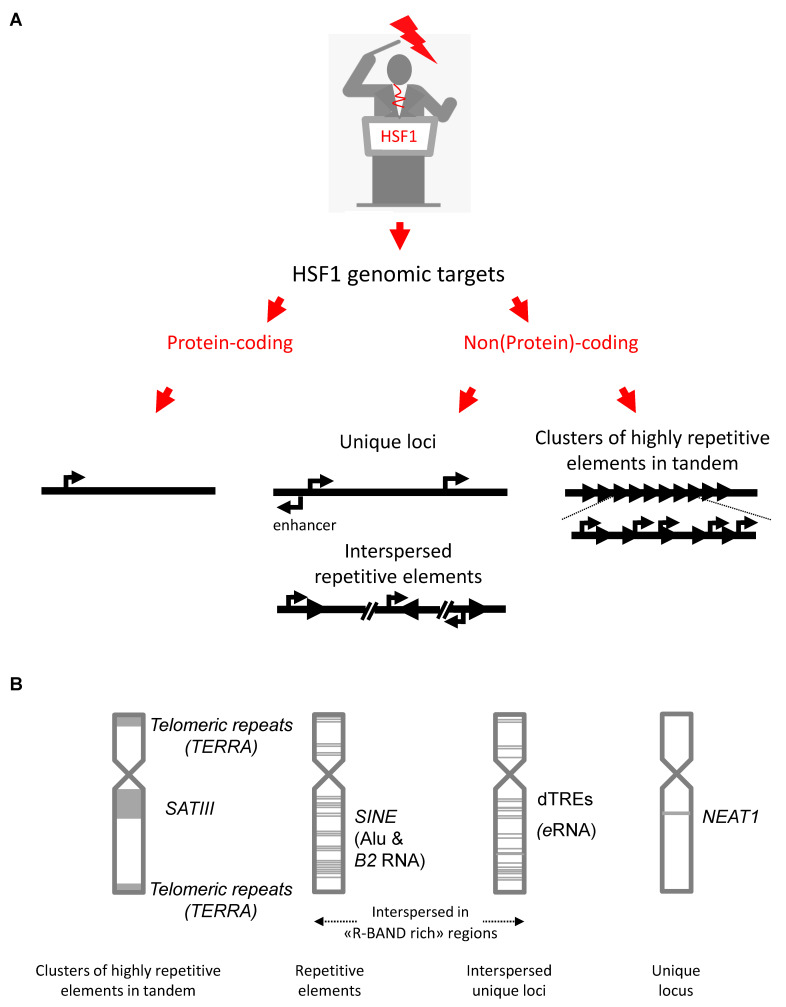
Genomic targets of the HSF1-mediated heat-shock response. (**A**) Several types of genomic sequences are targeted and transcribed by HSF1. They include protein-coding genes, such as HSP genes, and non(protein)-coding sequences. This later group includes non-repetitive elements present at a single locus (e.g., *NEAT1*, *distal Transcription Regulatory elements (dTREs) transcribed into eRNAs*) and repetitive interspersed elements, present as a thousand copies within gene-rich regions (e.g., *SINE*s). Finally, non-coding sequences also include clusters of highly repetitive elements in tandem such as *SATIII* and Telomeric repeats (transcribed into TERRA), located at pericentric and telomeric regions. (**B**) Distribution of non-coding sequences along human chromosomes. *SATIII* lncRNA and *TERRA* are transcribed from large regions located at pericentric and telomeric regions, respectively (primarily from chromosome 9 in the case of *SATIII* lncRNA). *SINE* sequences are mainly present in the gene-rich and GC-rich regions (R-bands). *eRNA*s are transcribed from the dTREs present upstream of the promoter and gene transcription start sites (TSS) and are, therefore, likely to be widespread and possibly more abundant in gene-rich regions. The *NEAT1* gene is present at a single locus.

**Figure 2 genes-13-00597-f002:**
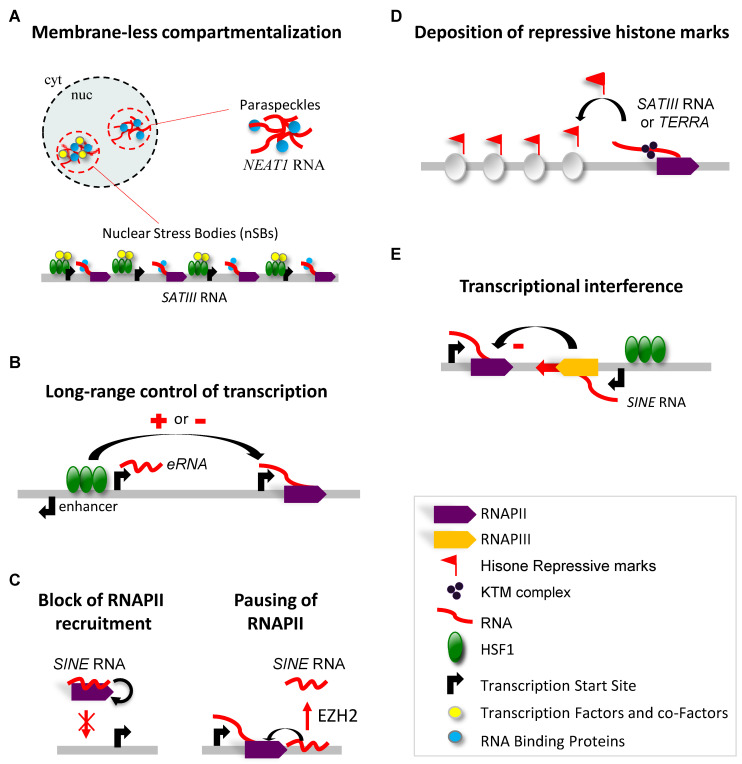
Molecular roles assigned to the non(protein)-coding RNAs in the heat shock response. (**A**) The ncRNAs produced at pericentric regions are thought to play a role in *cis* through the recruitment, and possible sequestration, of transcriptional repressive complexes. Transcriptional and co-transcriptional regulators as well as RNA-binding factors such as splicing factors accumulate at *SATIII* transcribed genomic sites upon HS. The resulting transient accumulation of transcription and splicing factors at these sites is thought to possibly form membrane-less compartments and cause a rapid and reversible depletion/concentration of these factors from the rest of the nucleus. Likewise, the nuclear accumulation of *NEAT1* lncRNA allows the formation of membrane-less nuclear structures known as paraspeckles, with a role in microRNA processing. (**B**) HSF1-dependent bi-directional transcription of *dTREs into eRNAs* may promote or repress transcription by RNAPII of protein-coding genes, upon heat shock. (**C**) *SINE* RNAs are thought to repress transcription by disrupting contacts between RNAPII and promoter DNA. In addition, specific *SINE* RNA-binding sites have been identified at many genic and intergenic targets proximal to RNAPII pausing, where they may prevent transcriptional elongation. (**D**) The lncRNAs produced at pericentric regions and telomeric regions are thought to play a role in *cis* through the recruitment, and possible sequestration, of transcriptional repressive complexes. (**E**) HSF1-dependent transcriptional activation of *SINE* sequences in an “antisense” orientation may also cause gene repression of the genes transcribed in a “sense” orientation.

**Table 1 genes-13-00597-t001:** List of ncRNAs that have their production activated by HSF1-induced transcription. Note: For genes and RNAs we follow the recommendation of Seal and coauthors [18].

ncRNA Production Activated by HSF1.	Length	Internal Repetitive Elements	Multiple Copies	HSE within Promoter Region	Genome Localization	Molecular Function in HS Cells
*SATIII*	from 2 kb to 5 kb and more	Yes (Tandem repeats, 5 b long)	Yes	Yes (in silico)	Multiple sites at heterochromatin pericentric regions	Titration of transcription factors
						Reorientation of splicing decision
						Maintenance of centromeric heterochromatin
						Maintenance of repressive histone marks
*Alu*	~280 b	No	Yes	Yes (ChIP)	Multiple and dispersed sites	RNAPII inhibition
						Impact of gene expression through antisens RNA
						Impact on transcriptional elongation
						Recruitment of repressive transcriptional complexes
						Alteration genome integrity through retrotransposition
*eRNA*	From 50 b to 2 kb	No	No	Yes (ChIP)	Multiple and dispersed sites	Control of gene expression
*NEAT1*	~3 kb (*NEAT1-1*) ~20 kb (*NEAT1-2*)	No	No	Yes (ChIP)	Unique locus	miRNA biogenesis
						HSP genes down regulation following HS
*TERRA*	From 100 b to less than 100 kb	Yes (Tandem repeats, 6 b unit)	Yes	Yes (ChIP)	Multiple sites at telomeres	Genome protection through telomere protection
						Telomeric heterochromatin reformation by recruiting repressive chromatin marks

**Table 2 genes-13-00597-t002:** General characteristics (sequence, size, chromosome distribution) of mouse and human satellite repetitive units of pericentric and pericentric origin.

Human		
centromeric repetitive motif ([20])		
*Minor SAT* (120 bp)	all chromosomes	GGAAAATGATAAAAACCACACTGTAGAACATATTAGATGAGTGAGTTACACTGAAAAACACATTCGTTGGAAACGGGATTTGTAGAACAGTGTATATCAATGAGTTACAATGAGAAACAT
pericentric repetitive motif ([21])		
*Major SAT* (234 bp)	all chromosomes	CCTGGAATATGGCGAGAAAACTGAAAATCACGGAAAATGAGAAATACACACTTTAGGACGTGAAATATGGCGAGGAAAACTGAAAAAGGTGGAAAATTTAGAAATGTCCACTGTAGGACGTGGAATATGGCAAGAAAACTGAAAATCATGGAAAATGAGAAACATCCACTTGACGACTTGAAAAATGACGAAATCACTAAAAAACGTGAAAAATGAGAAATGCACACTGAAGGA
**Mouse**		
centromeric repetitive motif ([19])		
*Alphoid* (171 bp)	all chromosomes	CTTCTGTCTAGTTTTTATATGAAGATATTCCCGTTTCCAACCAAGGCCTCAAAGCGGTCCAAATATCCACAAGCTGATTCTACAAAAAGAGTGTTTCAAAACTGCTCTATGAAAAGGAAGGTTCAACTCTGTGAGTTGAATGTATACATCACAAAGAAGTTTCTGAGAATG
pericentric satellite repetitive motif ([22,23,24,25,26,27])		
*SATI*	chrs 3, 4, 13, 14, 15, 21, 22, Y	Alternance of fragments A (17 bp) = ACATAAAATATG/CAAAGT and B (25 bp) B1: ACAT/CCCAAATATAG/TATTA/TTATA/TCTGT and B2: ACCCAAAGT/GCCATAT/GCATTA/CTATACT
*SATII*	chrs 1, 2, 7, 10, 15, 16, 17, 22	(CATTC)n degenerated
*SATIII*	chrs 1, 3, 5, 7, 9, 10, 17, Y, acrocentric	(CATTC)n

## Data Availability

Not applicable.

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
