# Peer review of "HSF1-Activated Non-Coding Stress Response: Satellite lncRNAs and Beyond, an Emerging Story with a Complex Scenario"

_genes, 2022, doi:10.3390/genes13040597_

Round 1

Reviewer 1 Report

In the review ‚HSF1-activated non-coding stress response: Satellite lncRNAs and beyond, an emerging story with a complex scenario’ the authors describe recent knowledge about HSF1 regulations. The review is very interesting and clearly written.

However, there are some parts which should be improved:

  • The part on coding responses of HSF1 should be extended such that the comparison to non-coding responses can be improved.
  • references are missing, e.g. Zhu et al 2018, Mendez-Bermudez 2018, Padeken 2019, Gaglia 2020, Yamazaki 2018, Huo, 2019…
  • other satellites (I,II,alpha) should be included
  • effects on genomic integrity in particular on R-loop formation should be included
  • authors should try to improve some sections, e.g. acetylation, splicing and so on. The descriptions are sometimes not very clear.

Minor points:

35-37: references are missing

45: The part ‘the biological relevance of this HSF1-mediated non-coding response is now clear…’.  is misleading, because the relevance of the upregulation of ncRNAs is by far not yet clear. Should be improved

71: ‘they’: the sentence does not make sense.

88: A figure on satellites would be helpful

91: H3K9me2 (Padeken), BRCA1 mediated ub? (Zhu 2011,2018), other makers (Saksouk 2015)

218: ..in cells submitted to stress conditions…

219: define mild stress

230: Reference 88 should be added.

262: why italic?

264: which role?

297: redundant to line 187ff

299: please be more specific about the ‘microprocessor complex’

305: an up-regulation

306/307: context of HSP? 

336: please add reference

360-367: the authors could be more specific on the mechanistic differences between developmental processes, on-stress conditions and upon stress recovery

374-387: should be extended since the comparison between species is an additional important aspect.  

412: evidence by Zhu et al 2018 - should be mentioned

449: effects on global and local replication (Mendez-Bermudez 2018)

473: SatIII associate with Toposiomerase II under stress and etoposide treatment and are involved in recruitment of Top2A (Kanne et al. 2021)

515: HSF1 foci are also subject to LLPS (Gaglia et al), similar reports for NEAT1 (Yamazaki et al.)  on paraspeckles and Major Satellites (Huo et al.)

The text should be improved and read by a native English speaker.  

Author Response

Thank you very much for your comments, which we think were indeed relevant. We hope the new version is now better.

Reviewer 1

In the review ‚HSF1-activated non-coding stress response: Satellite lncRNAs and beyond, an emerging story with a complex scenario’ the authors describe recent knowledge about HSF1 regulations. The review is very interesting and clearly written.

However, there are some parts which should be improved:

  • The part on coding responses of HSF1 should be extended such that the comparison to non-coding responses can be improved. A reference to the paper of Vihervaarat 2017 is now added.
  • references are missing, e.g. Zhu et al 2018, Mendez-Bermudez 2018, Padeken 2019, Gaglia 2020, Yamazaki 2018, Huo, 2019… These referenecs are now included
  • other satellites (I,II,alpha) should be included : See Table I (alpha sat, not activated by HSF1, were not included). Done (new table I included)
  • effects on genomic integrity in particular on R-loop formation should be included (a reference to R-loop has been added)
  • authors should try to improve some sections, e.g. acetylation, splicing and so on. The descriptions are sometimes not very clear. We hope that the modifications we have brought make these sections more clear.

Minor points:

35-37: references are missing :

Replace :

In addition, non-HSP protein-coding genes targeted by HSF1 have been identified for their anti-apoptotic roles, or for their role in the control of the cell cycle {7}.

with

In addition, non-HSP protein-coding genes targeted by HSF1 have been identified with role in the regulation of apoptosis, cellular defense to Reactive Oxygen Species (ROS), cell membrane and chromatin organization [7-9].

45: The part ‘the biological relevance of this HSF1-mediated non-coding response is now clear…’.  is misleading, because the relevance of the upregulation of ncRNAs is by far not yet clear. Should be improved

It is now replaced by

The biological relevance of this HSF1-mediated non-coding response, the mechanisms involving the regulatory lncRNAs produced are only starting to be understood.

71 : ‘they’: the sentence does not make sense.

‘They’ is now replaced by

‘these assumptions’

88: A figure on satellites would be helpful

Now included as Table 1 (Former Table 1 now being Table 2)

91: H3K9me2 (Padeken), BRCA1 mediated ub? (Zhu 2011, 2018), other makers (Saksouk 2015)

Volume 70, Issue 5, 7 June 2018, Pages 842-853.e7

Has been completed

Pericentric regions are enriched in methylated DNA, [29] and other repressive marks, such as H3K9me2/3 and Heterochromatin Protein 1 (HP1) [18, 30, 31]. In addition, repression at pericentric regions also involves a BRCA1 mediated H2A monoubiquitinylation [32, 33].

219: define mild stress

This is now notified as follows:

In human stressed cells, SATIII lncRNAs remain at their site of transcription, even several hours after mild stress exposure (30 min exposure at 43°C)

230: Reference 88 should be added.

Indeed, ref 83 (Hussong et al. 2017) is now added

262: why italic?

Note: For human SATIII lncRNA and NEAT1 RNAs, we follow the recommendation of Seal and coauthors [169]. The names of other transcripts are given according to what we found in the literature.

264: which role?

Stress induced dephosphorylation of the serine- and arginine-rich protein SRSF10 (SRP38), thought to prevent its association with pre-mRNA [80] plays an important role in the heat-induced inhibition of splicing [81].

Is completed with  :

Stress induced dephosphorylation of the SRSF10 (SRP38), plays an important role in the heat-induced inhibition of splicing [94]. Dephosphorylated serine- and arginine-rich protein SRSF10 (SRP38) binds to U1 snRNP-associated protein U1 70K, and prevents the interaction of U1 70K with other SR proteins [95].

297: redundant to line 187ff

Indeed.

Now the description is in paragraph 3.2 line 314.

Like SATIII, NEAT1 lncRNAs also seem to play a role in RNA processing. Indeed, NEAT1 lncRNAs have been found to anchor the DROSHA-DGCR8 microprocessor complex to paraspeckles via a pri-miR-612 (and possibly additional) stem-loop structure(s) present within NEAT1 sequence [103].

299: please be more specific about the ‘microprocessor complex’

Line 314

Indeed, NEAT1 lncRNAs have been found to anchor the DROSHA-DGCR8 microprocessor complex to paraspeckles via a pri-miR-612 (and possibly additional) stem-loop structure(s) present within NEAT1 sequence [103].

305: an up-regulation

‘an up-regulation’ now replaces ‘a up-regulation’

306/307: context of HSP? 

Interestingly, a reduction of nearly all expressed miRNAs is observed in cells lacking NEAT1 [90]. In this general context, a up-regulation of miRNAs may possibly result from the stress-induced accumulation of NEAT1 already reported in heat-shocked cells [64] (Figure 2A). An up-regulation of HSP genes has been reported in stressed cells knock down for NEAT1. However, the mechanisms involved have not been identified yet [64].

Is now replaced by

Indeed, a reduction of almost all expressed miRNAs is observed in cells lacking NEAT1 [90] and could be related to the up-regulation of HSP genes, reported in stressed cells knocked down for NEAT1 [64] (Figure 2A). The exact implication of NEAT1 up-regulation on miRNA mediated changes in gene expression in heat-shocked cells, however, remains to be determined.

336: please add reference

The missing references are now included in the text.

360-367: the authors could be more specific on the mechanistic differences between developmental processes, on-stress conditions and upon stress recovery

This section has been completed. See paragraph 3.4

374-387: should be extended since the comparison between species is an additional important aspect.  

This section is now completed and improved..

412: evidence by Zhu et al 2018 - should be mentioned

Line 462 ref 144

Overexpression of SATIII RNAs in breast cancer deficient for BRCA1 has been found to destabilize replication forks through the interaction of SATIII RNA with the members of the complex required for DNA replication composed of BRCA1 interacting protein [144]. The titration of members of the BRCA1 complex away from stalled replication fork results in the formation of RNA-DNA hybrids and in the accumulation of gH2AX (a marker for DNA damage) at replication fork [144].

449: effects on global and local replication (Mendez-Bermudez 2018)

Ref 148 line 492

Moreover, TRF2 prevents telomeres from end-fusions elicited by the non-homologous end-joining pathway [53] and assists replication fork progression through pericentric regions, by preventing the accumulation of G-quadruplexes-like structures [148].

473: SatIII associate with Topoisomerase II under stress and etoposide treatment and are involved in recruitment of Top2A (Kanne et al. 2021)

This reference is now added (45) line 469

Finally, SATIII lncRNAs appear as interesting therapeutic markers in cancer therapy. Together with heat-shocked cells, cancer cells overexpressing SATIII RNA as a consequence of DNA hypomethylation at pericentric heterochromatic regions, are resistant to etoposide, a topoisomerase IIa (TOP2a) inhibitor. Interaction of TOP2a with SATIII RNA at nSBs protects TOP2A from interaction with etoposide, preventing increased DNA damages in etoposide treated cells [45].

515: HSF1 foci are also subject to LLPS (Gaglia et al), similar reports for NEAT1 (Yamazaki et al.)  on paraspeckles and Major Satellites (Huo et al.)

These three references have been inserted line 554 (162,163,164)

Finally, growing attention is also being paid to the contribution of lncRNAs in nucleating, maintaining and regulating the formation of membrane-less nuclear compartments through liquid-liquid phase separations (LLPS) [160-162]. Solidification of HSF1 foci upon sustained stressing environment was found to have negative impact on HSP gene expression [163]. Trapping of HSF1 within nSBs has been proposed to mark the cells with excessive proteotoxic damage and turn a reversible response into a signal to apoptosis with irreversible outcome [163]. Other HSF1-dependent stress-induced non-coding transcription, such as NEAT1 transcription [164] may serve as effective entities to orchestrate and reshape gene expression.

The text should be improved and read by a native English speaker.

Done.  

Reviewer 2 Report

This review article provides a welcome update and synthesis of the roles and functions of HSF1-induced long and non-coding RNA. The article is clear and structured, although the sectioning hierarchy can, in my opinion, be improved. This review article largely covers the knowledge on the subject, however it is regrettable that the recent references are few (7 articles cited out of 154 for the period 2019 -2021 ) while it is an active research topic with scientific news.

Minor corrections:

1) Include more recent references (over the last three years) to ensure a longer shelf life for this review article

2) Revisit the graphic design of figure 2 which as it stands does not bring any additional precision to the text and is, in my opinion, not independently readable.

3) Review the sectioning hierarchy. The following proposal is only indicative:
1 - Introduction
2 - The Genomic Noncoding Sequences Transcribed Under the Direct Control of HSF1 (1.1 in actual form)
3 - Molecular functions of HSF1-driven production of lncRNAs in response to heat stress (1.2 in actual form)
4 - Conclusions (2 in actual form)

Author Response

We would like to thank reviewer 2.
The requested changes have been made. We believe they indeed substantially improve the quality of the review. 

Reviewer 2

This review article provides a welcome update and synthesis of the roles and functions of HSF1-induced long and non-coding RNA. The article is clear and structured, although the sectioning hierarchy can, in my opinion, be improved. This review article largely covers the knowledge on the subject, however it is regrettable that the recent references are few (7 articles cited out of 154 for the period 2019 -2021 ) while it is an active research topic with scientific news.

Minor corrections:

1) Include more recent references (over the last three years) to ensure a longer shelf life for this review article

The following references have been added.

Hoyt, S. J., Storer, J. M., Hartley, G. A., Grady, P. G., Gershman, A., de Lima, L. G., ... & O’Neill, R. J. (2021). From telomere to telomere: the transcriptional and epigenetic state of human repeat elements. bioRxiv.

Nurk, S., Koren, S., Rhie, A., Rautiainen, M., Bzikadze, A. V., Mikheenko, A., ... & Phillippy, A. M. (2021). The complete sequence of a human genome. bioRxiv.

Saksouk, N., Barth, T. K., Ziegler-Birling, C., Olova, N., Nowak, A., Rey, E., ... & Déjardin, J. (2014). Redundant mechanisms to form silent chromatin at pericentromeric regions rely on BEND3 and DNA methylation. Molecular cell, 56(4), 580-594.

Peters, A. H., O'Carroll, D., Scherthan, H., Mechtler, K., Sauer, S., Schöfer, C., ... & Jenuwein, T. (2001). Loss of the Suv39h histone methyltransferases impairs mammalian heterochromatin and genome stability. Cell, 107(3), 323-337.

Ninomiya, K., Iwakiri, J., Aly, M. K., Sakaguchi, Y., Adachi, S., Natsume, T., ... & Hirose, T. (2021). m6A modification of HSATIII lncRNAs regulates temperature‐dependent splicing. The EMBO Journal, 40(15), e107976.

Xian Ma, Y., Fan, S., Xiong, J., Yuan, R. Q., Meng, Q., Gao, M., ... & Rosen, E. M. (2003). Role of BRCA1 in heat shock response. Oncogene, 22(1), 10-27.

Zhu, Q., Hoong, N., Aslanian, A., Hara, T., Benner, C., Heinz, S., ... & Verma, I. M. (2018). Heterochromatin-encoded satellite RNAs induce breast cancer. Molecular cell, 70(5), 842-853.

Gaglia, G., Rashid, R., Yapp, C., Joshi, G. N., Li, C. G., Lindquist, S. L., ... & Santagata, S. (2020). HSF1 phase transition mediates stress adaptation and cell fate decisions. Nature cell biology, 22(2), 151-158.

Zhu, Q., Pao, G. M., Huynh, A. M., Suh, H., Tonnu, N., Nederlof, P. M., ... & Verma, I. M. (2011). BRCA1 tumour suppression occurs via heterochromatin-mediated silencing. Nature, 477(7363), 179-184.

Kanne, J., Hussong, M., Isensee, J., Muñoz-López, Á., Wolffgramm, J., Heß, F., ... & Schweiger, M. R. (2021). Pericentromeric Satellite III transcripts induce etoposide resistance. Cell death & disease, 12(6), 1-15.

Gutbrod, M. J., Roche, B., Steinberg, J. I., Lakhani, A. A., Chang, K., Schorn, A. J., & Martienssen, R. A. (2022). Dicer promotes genome stability via the bromodomain transcriptional co-activator BRD4. Nature Communications, 13(1), 1-14.

Mendez-Bermudez, A., Lototska, L., Bauwens, S., Giraud-Panis, M. J., Croce, O., Jamet, K., ... & Ye, J. (2018). Genome-wide control of heterochromatin replication by the telomere capping protein TRF2. Molecular cell, 70(3), 449-461.

Yamazaki, T., Souquere, S., Chujo, T., Kobelke, S., Chong, Y. S., Fox, A. H., ... & Hirose, T. (2018). Functional domains of NEAT1 architectural lncRNA induce paraspeckle assembly through phase separation. Molecular cell, 70(6), 1038-1053.

Huo, X., Ji, L., Zhang, Y., Lv, P., Cao, X., Wang, Q., ... & Wen, B. (2020). The nuclear matrix protein SAFB cooperates with major satellite RNAs to stabilize heterochromatin architecture partially through phase separation. Molecular Cell, 77(2), 368-383.

Altemose, N. A Classical Revival: Human Satellite DNAs Enter The Genomics Era. Preprints 2022, 2022020009 (doi: 10.20944/preprints202202.0009.v1). Altemose, N. A Classical Revival: Human Satellite DNAs Enter The Genomics Era. Preprints 2022, 2022020009 (doi: 10.20944/preprints202202.0009.v1).

2) Revisit the graphic design of figure 2 which as it stands does not bring any additional precision to the text and is, in my opinion, not independently readable.

done

3) Review the sectioning hierarchy. The following proposal is only indicative:
1 - Introduction
2 - The Genomic Noncoding Sequences Transcribed Under the Direct Control of HSF1 (1.1 in actual form)
3 - Molecular functions of HSF1-driven production of lncRNAs in response to heat stress (1.2 in actual form)
4 - Conclusions (2 in actual form)

done